# Chimeric Antigen Receptor T-Cell Therapy and Hematopoiesis

**DOI:** 10.3390/cells12040531

**Published:** 2023-02-07

**Authors:** Bryanna Reinhardt, Patrick Lee, Joshua P. Sasine

**Affiliations:** 1School of Medicine, Louisiana State University Health Sciences Center, New Orleans, LA 70112, USA; 2Department of Medicine, Cedars-Sinai Medical Center, Los Angeles, CA 90048, USA; 3Department of Medicine, Division of Hematology and Cellular Therapy, Samuel Oschin Cancer Center, Cedars-Sinai Medical Center, Los Angeles, CA 90048, USA

**Keywords:** CAR T, hematopoietic stem cells, cytopenia, cytokine release syndrome, inflammatory toxicity, bone marrow failure, conditioning, clonal hematopoiesis

## Abstract

Chimeric Antigen Receptor (CAR) T-cell therapy is a promising treatment option for patients suffering from B-cell- and plasma cell-derived hematologic malignancies and is being adapted for the treatment of solid cancers. However, CAR T is associated with frequently severe toxicities such as cytokine release syndrome (CRS), immune effector cell-associated neurotoxicity syndrome (ICANS), macrophage activation syndrome (MAS), and prolonged cytopenias—a reduction in the number of mature blood cells of one or more lineage. Although we understand some drivers of these toxicities, their mechanisms remain under investigation. Since the CAR T regimen is a complex, multi-step process with frequent adverse events, ways to improve the benefit-to-risk ratio are needed. In this review, we discuss a variety of potential solutions being investigated to address the limitations of CAR T. First, we discuss the incidence and characteristics of CAR T-related cytopenias and their association with reduced CAR T-cell efficacy. We review approaches to managing or mitigating cytopenias during the CAR T regimen—including the use of growth factors, allogeneic rescue, autologous hematopoietic stem cell infusion, and alternative conditioning regimens. Finally, we introduce novel methods to improve CAR T-cell-infusion products and the implications of CAR T and clonal hematopoiesis.

## 1. Introduction

Autologous Chimeric Antigen Receptor (CAR) T-cell therapy is approved by the FDA to treat several relapsed or refractory (r/r) hematological malignancies such as Diffuse Large B-Cell Lymphoma (DLBCL), B-cell Acute Lymphoblastic Leukemia (ALL), Follicular Lymphoma (FL), Mantle Cell Lymphoma (MCL), and Multiple Myeloma (MM). Much of the enthusiasm for CAR T is supported by the relatively high response rates and durations for these historically difficult-to-treat patient populations. Remarkably, response rates to CAR T-cell therapy in patients with r/r disease are significantly better than alternatives [1], although durability to treatment is varied [2]. For example, typical complete remission (CR) rates to standard immunochemotherapy for r/r diffuse large B-cell lymphoma (DLBCL) can be as low as 7% [3] compared to those for CAR T which reached 58% in the ZUMA-1 trial [4]. In an analysis comparing B-cell lymphoma patient outcomes two years after CAR T or salvage therapy, Sattva et al. found that patients who received axi-cel in the ZUMA-1 trial had a significantly higher ORR and 73% reduced risk of death compared to standard salvage therapy [5]. On the other hand, an average of 60% of patients treated with CAR T for DLBCL experienced disease progression, potentially relating to the composition of the CAR T infusion product and proportion of CAR T regulatory (CAR Treg) cells [2]. 

Optimizing CAR T efficacy and minimizing toxicity are important next steps to improving patient outcomes and designating CAR T as an earlier phase therapy for patients with hematologic malignancies. In addition to well-characterized CAR T toxicities—such as cytokine release syndrome (CRS), immune effector cell-associated neurotoxicity syndrome (ICANS), and CAR T-associated cytopenias (low blood counts)—there are increasingly recognized phenomena that have implications for overall prognosis, CAR T efficacy, and patient experience. 

The mechanism driving prolonged CAR T-associated cytopenias is still unknown. The conditioning chemotherapy certainly has an impact; however, it is not a sufficient explanation for the degree of long-term cytopenias in many patients after CAR T. Using the FCR regimen (fludarabine, cyclophosphamide, and rituximab) to treat patients with chronic lymphocytic leukemia (CLL), the rate of prolonged cytopenias in fit patients is only 4.6% [6]. The dose and schedule of fludarabine and cyclophosphamide is like that used in patients with CAR T. Therefore, it is likely that other factors—such as pre-treatment bone marrow health, baseline inflammation, and inflammation driven by CAR T—could contribute to the prolonged cytopenias after CAR T. Other, yet unknown mechanisms are also possible.

In this review, we summarize the characteristics and risk factors associated with post CAR T-cell cytopenias, as well as the current evidence linking cytopenias and therapeutic response. We then delve into ongoing avenues of investigation to mitigate these cytopenias, including both modifications to the CAR T-cell regimen and adjunctive therapies.

## 2. Incidence and Characteristics of CAR T-Associated Cytopenias

CAR T is usually associated with cytopenias. These are often biphasic [7] and sometimes prolonged over several months. Cytopenias often lead to infections, need for transfusions of blood products, and increased CAR T morbidity. While CRS and ICANS can be treated with immune suppressing agents such the IL-6 monoclonal antibody, tocilizumab, and corticosteroids, cytopenias are more difficult to address. In the long run, cytopenias are particularly problematic because further treatment is often hindered or prohibited for patients who experience a relapse due to cytopenias. Most treatment for hematologic malignancies induces hematopoietic toxicity and the ability to safely deliver the therapy relies on intact hematopoiesis.

The severity and duration of post-infusion cytopenias are varied across patients, and generally include incidences of neutropenia, thrombocytopenia, and anemia which reached rates of 78%, 38%, and 43%, respectively, in the ZUMA-1 trial assessing axicabtagene ciloleucel (axi-cel) for r/r DLBCL [8]. In the JULIET trial, which assessed tisagenlecleucel (tisa-cel) in r/r DLBCL, the average rates of anemia, neutropenia, and thrombocytopenia were 40%, 20%, and 13%, respectively [9]. Moreover, Sharma et al. reported that patients receiving axi-cel, tisa-cel lisocabtagene maraleucel (liso-cel), or locally produced CAR T with a 4-1BB or CD28 co-stimulatory domain across 10 studies, experienced neutropenia, thrombocytopenia, and anemia at a rate of at least 9%, 14%, and 16% by day 28 post CAR T [10]. This suggests that the CAR product itself, chemotherapy conditioning regimen, and disease being treated are important variables in determining the risk of cytopenias.

Cytopenias are also common in real-world datasets. In a retrospective analysis of DLBCL patients receiving tisa-cel or axi-cel, persistent neutropenia was observed in 9% of patients (median duration 9 days; *n* = 31; range: 2–205), persistent thrombocytopenia in 65% of patients (median duration 95.5 days), and persistent anemia in 72% of patients (median duration 125 days; *n* = 31; range: 9–303) [11]. In this study, neutropenia after CAR T was less common than thrombocytopenia and even less common than anemia, which was the most frequent. These cytopenias are common and frequently last months. Patients often require regular transfusions and are at increased risk for infections.

Prolonged cytopenias are common for BCMA-directed CAR T as well. Grade 3 or 4 cytopenia at day 0, 60, 120, 180, and 360 post-CAR T was 39%, 33%, 28%, 13%, and 7% respectively in one study of 90 patients with multiple myeloma receiving BCMA CAR T [12]. Prolonged cytopenias are a risk associated with CAR T therapy as a class; they are not specific to CD19-directed CAR T. As the number of antigen targets for CAR T grows, we expect prolonged cytopenias to be an antigen-independent risk that pervades the entire class.

## 3. Factors Associated with Post-CAR T Cytopenias

Not surprisingly, some studies have shown that baseline cytopenias are associated with prolonged cytopenias post CAR T [9]. Many patients receiving CAR T have been heavily pre-treated with cytotoxic chemotherapy which can damage the hematopoietic stem cells and the bone marrow microenvironment durability.

Systemic inflammation driven by increased levels of immune mediators promotes hematopoietic stem cell exhaustion [13] and this inflammation, especially following genotoxic CAR T conditioning treatment, can cause severe and long-lasting cytopenias. This phenomenon can be particularly problematic given the high incidence of inflammatory complications of CAR T-cell therapy.

Pretreatment hematopoietic reserve and inflammatory states appear to be important factors associated with post CAR T-cell cytopenias and their duration. A prolonged cytopenia is defined as an episode of lower-than-normal blood counts that persist for over two weeks to three months without evidence of significant hematopoietic recovery [14]. In a retrospective analysis of 258 patients treated with axi-cel or tisa-cel, pre-treatment low platelet counts, absolute neutrophil count (ANC), and hemoglobin as well as high C-reactive protein (CRP) and ferritin were significantly associated with delayed cytopenias after CAR T [15]. These findings formed the basis of the CAR-HEMATOTOX model, which combines baseline biomarkers (prior to conditioning chemotherapy) into one score that can be used as a guide for predicting patients’ hematopoietic recovery following CAR T. Other studies have also shown that pre-conditioning platelet counts are associated with post-infusion thrombocytopenia [16,17].

In a retrospective analysis of 133 patients with r/r lymphoma receiving CAR T, early cytopenia was associated with peak IL-6, CRP, and ferritin levels as well as incidence and severity of CRS, illustrating that management of baseline inflammation and inflammatory toxicities may decrease incidence and severity of post-infusion cytopenias [16]. In a different analysis of 83 patients treated with axi-cel, tisa-cel, and the CD19-28z CAR T-cell therapy for ALL, within the first week, hemoglobin nadir was 7.1 g/dl, platelets were 29.5 × 10^3^/μL, ANC was 0, and WBC was 0.2 × 10^3^/μL [18]. Recovery of hemoglobin, platelets, neutrophils, and WBC by one year post CAR T therapy was observed in 67%, 78%, 89%, and 89% of patients, respectively. Thus, a significant proportion of patients experienced prolonged cyopenias after treatment. Recovery of cell numbers by one month was significantly associated with baseline cytopenias, as was the CAR construct (tisa-cel was associated with better cell hematopoietic recovery). Increasing grade of CRS or ICANS was negatively associated with hematopoietic recovery, supporting the notion that CAR T-associated inflammation is detrimental to hematopoietic stem/progenitor function.

One potential explanation for the persistent cytopenias post CAR T is emergence of clonal hematopoiesis or myelodysplastic syndrome which may develop after several lines of genotoxic chemotherapy regimens [19]. In the previous study of 83 ALL patients [18], there was only one occurrence of myelodysplasia (MDS) following a patient’s relapse to CAR T, although other studies have reported higher incidences of MDS development [20]. Specifically, in a report of late events in 86 patients following CAR T treatment, 5% developed MDS and 16% experienced prolonged cytopenias [20]. Three of 19 patients in CR who did not have MDS experienced prolonged cytopenias for an average of 18.45 months following CAR T therapy [20]. While the development of MDS post CAR T is associated with cytopenias, the occurrence of prolonged cytopenias is not usually due to MDS. Clonal myeloid disorders are insufficient to explain most CAR T patients with prolonged cytopenias. Rather, cytopenias correlate with additional factors such as pre-lymphodepletion cell numbers and previous lines of therapy [4,21].

Post-infusion cytopenias also increase infection risk, largely due to the decrease in leukocytes available to fight off pathogens. However, repeated transfusion of red blood cells or platelets can increase infection risk as well. Neutropenia may result from a combination of lymphodepleting chemotherapy and immune dysregulation resulting from CRS and ICANS [22]. Multiple studies have correlated severe inflammatory toxicity with infections in ALL and B-Cell Non-Hodgkin Lymphoma patients [22,23]. Additionally, the increased infection risk may be related to age [24], previous infection prior to CAR T [25], the underlying hematologic malignancy [24], severity of CRS and ICANS (which are also risk factors for prolonged cytopenias following CAR T) [18,26].

CAR T-cell therapy can induce prolonged hematologic toxicity (PHT), which is further defined as incidence of grade > 3 neutropenia or thrombocytopenia beyond 29 days following CAR T infusion [27]. Nagle et al. reported that 58% of adult patients with r/r DLBCL treated with CAR-T between 2018 and 2020 developed PHT, which was associated with a 45% decrease in overall survival [27]. Risk factors associated with PHT include CRS, treatment with tocilizumab, administration of steroids, peak ferritin > 5000 ng/mL, and peak CRP > 100 mg/L. Mitigating cytopenias can be a useful first step in preventing the development of PHT.

The severity of CRS and lower pre-conditioning platelet count are predictive of hematologic toxicity in patients receiving CD19 CAR T-cell therapy [17]. In a retrospective analysis of 83 patients receiving axi-cel, tisa-cel, CD19-28z CAR T for B-ALL, or B-cell maturation antigen targeting CAR T for MM, Jain et al. found that the patients who had not progressed or died recovered hemoglobin, platelet, neutrophil, and white blood cell counts [18]. Additionally, increased severity of CRS and ICANS were associated with decreased likelihood of hematopoietic recovery at one month after CAR T [18]. Hematopoietic stem and progenitor cells are vulnerable to systemic inflammation which affects their recovery [13] and inflammation associated with CAR T can damage hematopoietic stem and progenitor cells.

Multiple independent groups have reported evidence of non-CAR T-cells playing a role in prolonged cytopenias. Li et al. [28] used scRNA-sequencing on the bone marrow aspirates from 16 patients with DLBCL treated with axi-cel, in which 11 patients had grade 3–4 cytopenia at day 30 and 5 patients did not; 5 healthy controls were included as comparisons. They found an enrichment of GZMH+ FGFBP2+ CD8 T cells which did not express the CAR within patients with CAR T-associated cytopenias. The most expanded TCR clones were enriched within this population of CD8 T-cells, and these were significantly enriched for interferon (IFN) signaling. IFN gamma can impair self-renewal and differentiation of HSPCs but can be targeted using IFNG-neutralizing antibodies or eltrombopag.

Rejeski et al. [29] found similar results in a single patient with DLBCL treated with tisa-cel. After severe CRS and several episodes of infection, the patient developed prolonged pancytopenia. Using scRNA-sequencing, they also found an oligoclonal population of T cells which did not express the CAR gene but were CD8+ CD57+. The oligoclonal expansion and immunophenotype was strikingly like that seen in aplastic anemia and T-cell large granular lymphocytic leukemia, also associated with pancytopenia [16,17,19,23].

Future studies will clarify the mechanism of CAR T cytopenias with the hope that this will reveal a target for therapeutic intervention. One consistent theme is that inflammation associated with CAR T is one mechanism responsible for prolonged cytopenias.

## 4. Pre-Treatment Cytopenias and Reduced CAR T Efficacy

In addition to the direct health risks associated with cytopenias such as infection [30] and bleeding, cytopenias are associated with a decreased efficacy of CAR T-cell therapy. In a retrospective cohort analysis of patients receiving axi-cel for r/r DLBCL, the severity or duration of post-infusion cytopenias were associated with CAR T failure [31]. Moreover, as with other studies previously mentioned, there was a significant association between cytopenias and higher incidence and severity of CRS and ICANS.

In a study of 53 patients receiving CD19 CAR T for B-ALL, Hay et al. found that EFS and OS were significantly higher in patients who achieved minimal residual disease (MRD)-negative CR compared to those who did not, and MRD-negative CR was associated with patients having lower pre-conditioning lactate dehydrogenase concentration and higher pre-lymphodepletion platelet count [32].

Moreover, patients with a lower tumor burden [33], higher pre-conditioning cell counts [17,32], and quicker hematopoietic recovery post conditioning chemotherapy [18] tend to experience more benefit from CAR T, highlighting the importance of bone marrow health and recovering hematopoiesis in predicting CAR T efficacy.

On the other hand, increasing the intensity of conditioning chemotherapy prior to CAR T has consistently been one of the most important correlates of improved CAR T outcome [32]. This is because increased conditioning doses of fludarabine and cyclophosphamide correlate with CAR T expansion and efficacy. However, these increased doses, in turn, worsen the risk and duration of cytopenias. Given that, it is surprising to learn that cytopenias are associated with reduced CAR T efficacy. One might assume (incorrectly) that prolonged cytopenias could be a marker of deeper effects of conditioning chemotherapy and that patients with cytopenias might have a better efficacy from CAR T. Indeed, the mechanism of the benefit to conditioning chemotherapy is thought to be in part due to depletion of the cells themselves.

The causality of the association between cytopenias and reduced CAR T efficacy is still unknown. One hypothesis is that post-infusion cytopenias reflect general hematopoietic health, including T cell health. If so, post-CAR T cytopenias might only be a useful biomarker of CAR T efficacy. However, immunity functions as a collaborative effort in general, so a causal, actionable relationship is worth exploring. Multiple independent groups have found a role for macrophages in CAR T toxicity [34,35,36,37] and a recent animal study using CAR T in an acute myeloid leukemia model supported the notion that normal myeloid cells are critical for the durability of CAR T responses [38]. It is possible that augmenting hematopoietic recovery after CAR T could enhance the durability of responses.

One important exception to the above is the absolute lymphocyte count. In a retrospective study of NHL patients receiving CAR T, higher peak absolute lymphocyte count (ALC) expansion was significantly associated with CAR T levels in blood and durable CR [39]. Of note, the incidence and duration of CRS or ICANS were not associated with increased peak ALC, though direct measurement of CAR T expansion in other studies correlated with inflammatory toxicity [40].

## 5. Implications of Age-Associated Inflammation on Bone Marrow and CAR T Outcomes

Aging is associated with increased bone marrow inflammation and, as a result, age may influence patient response to CAR T therapy. Hematopoietic stem and progenitor cells (HSPCs) and stromal cells making up the bone marrow microenvironment are sensitive to hematopoietic stress signals that tend to increase with age. Bone marrow stress induces cell cycling which damages HSPCs.

Age also appears to increase the risk of infection post CAR T-cell therapy [24,25]. Single cell studies comparing niche cells from young and old mice revealed that older cells upregulate inflammatory responses and alter their differentiation patterns [41]. Inhibiting IL-1 signaling helped older niche cells recover from inflammatory stress, indicating that cytokines play an important role in mediating this process [41]. It is possible that CAR T-cell therapy, which induces cytokine release and bone marrow inflammation, may accelerate HSPC aging and increase the risk of bone marrow failure. This inflammation is characterized by CAR T-cell activation and subsequent secretion of inflammatory factors such as GM-CSF. In turn, these inflammatory mediators activate myeloid cells which induce the production and release of cytokines such as IL-6 and IL-1beta [42]. Hay et al. analyzed the kinetics and biomarkers of severe CRS and found that certain baseline characteristics—such as the use of Flu/Cy conditioning, high marrow tumor burden, higher CAR T dose, baseline thrombocytopenia, and CD8+ memory T-cell deficient CAR T infusion products—are independent predictors of CRS [43]. Since pre-treatment bone marrow health and cell counts are related to CAR T toxicity, age related bone marrow inflammation may exacerbate these side effects.

It remains unclear whether a patient’s age increases the risk of prolonged post-infusion cytopenias due to fludarabine and cyclophosphamide. Looking at the effects of chemotherapy alone, Tam et al. evaluated the long-term results of combination therapy with FCR in CLL patients and reported that post-therapy cytopenias were not significantly associated with age or performance status [44]. However, Strati et al. found that age was a risk factor for the development of prolonged cytopenias after FCR [45].

Another study reported that older patients (>65 years of age) with advanced NHL experienced no worse toxicity than younger patients receiving CAR T-cell therapy in the ZUMA-1 trial [46]. Additionally, Memorial Sloan Kettering Cancer Center reported that there were no significant differences in progression-free survival or post-relapse overall survival between older (>65 years of age) and younger patients among the 49 who received CAR T-cell therapy at their center between 2018 and 2019 [47]. However, the optimal threshold for defining younger and older patients remains unknown.

The anti-CD19 CAR T product with a 4-1BB co-stimulatory domain, tisa-cel, is approved for treating children with acute lymphoblastic leukemia. In the multicenter, phase I/II ELIANA trial, out of 75 children and young adults under ages 25 years old who were treated with tisa-cel, 81% achieved remission at three months post infusion. Out of this cohort, 50% achieved EFS and 76% OS at 12 months post infusion [48]. In a long-term follow-up analysis of adults receiving tisa-cel for ALL, Park et al. reported that 83% of patients achieved CR, with a median EFS of 6.1 months and median OS of 12.9 months [49]. Interestingly, response rates appeared similar across these different cohorts, although the younger patients experienced a more sustained response to CAR T, on average [50]. Interestingly, the incidence of severe CRS was higher at 37% in the ELIANA trial while it was 26% in the long-term follow-up study.

In a phase II, single-cohort, 25-center study of tisa-cel in pediatric and young adult patients with r/r ALL, Maude et al. found that by day 28, 41% of patients had unresolved, grade 3–4 thrombocytopenia and 53% of the patients had unresolved, grade 3–4 neutropenia. Nearly 50% of the patients received care in an intensive care unit, and 18 of the 40 patients with grade 3 or 4 neutropenia experienced grade 3 or 4 infections that were rarely associated with grade 3 human herpesvirus 6 (HHV-6) encephalitis or fatal encephalitis and systemic mycosis [48].

Although an aged bone marrow may have less reserve, clearly young patients are still at risk for severe cytopenias. Future clinical trials and studies assessing CAR T efficacy in children and adults, as well as research on bone marrow inflammation and microenvironment changes throughout CAR T treatment, may help clarify the mechanism and interplay of aging in this context.

## 6. Approaches to Mitigating and Managing Cytopenias

### 6.1. Paracrine Factor Modulation

As shown in Figure 1, there are several approaches to optimize the CAR T-cell therapy regimen, including paracrine factor modulation. Granulocyte-colony stimulating factor (G-CSF) has been used to increase granulocytes after patients receive CAR T [51]. G-CSF induces granulocyte production from hematopoietic progenitor cells and is used to help prevent infections during neutropenic episodes associated with severely myelosuppressive chemotherapy regimens and hematopoietic cell transplants [52]. A study in rodents suggested that myeloid growth factors may interact with CAR T cells in vivo, potentially worsening inflammatory toxicities [38]. Preclinical data that support this finding, suggesting that the downstream products of G-CSF-stimulated myeloid cells—such as IL-6 and IL-10—are largely responsible for the development of CRS upon T cell activation and expansion in vivo [37].

Clinical studies have not been consistent on the relationship of G-CSF with CAR T inflammatory toxicities. A retrospective study assessing the impacts of G-CSF administration after CAR T infusion reported that while G-CSF was not significantly associated with the incidence of CRS, it was associated with increased severity of CRS in patients with DLBCL [31]. In another study where G-CSF was administered to 35/70 patients receiving axi-cel for DLBCL, the severity and incidences of CRS was consistent across both groups, although the duration of CRS was significantly longer in the group of patients who received G-CSF (4.5 days versus 8 days) [53]. In both studies, the total duration of neutropenia was significantly lower in patients who received G-CSF compared to those who did not.

Opposing evidence from subsequent studies suggest that G-CSF could be safe after CAR T. Galli et al. reported that patients who received G-CSF starting at five days post-CAR T were not at increased risk of experiencing severe CRS or ICANS [54].

Multiple retrospective studies have shown that patients who receive G-CSF after CAR T have a poorer response to CAR T with a worsened overall survival [31,55]. The causality of this association is unknown. Stimulation of myeloid suppressor cells by G-CSF is possible, though other confounders exist, such as the cytopenias themselves, as discussed in the previous section. However, other studies have not shown an association between CAR T response and G-CSF use [51,54].

Additionally, G-CSF administration does not appear to significantly reduce the incidence or severity of infections, although it has been reported to reduce the duration of CAR T hospital stays in some studies [54,56,57]. Although most studies did not report a decrease in febrile neutropenia in this context, one report showed that patients who received early G-CSF had a significantly decreased incidence of febrile neutropenia (58%) compared to patients who did not receive early G-CSF or who received late G-CSF (81%) [51].

Taken together across studies, these results suggest that G-CSF can be useful for treating neutropenia but may have a harmful effect on inflammatory toxicities without a significant reduction in infection risk. The European Society for Blood and Marrow Transplantation recommended using G-CSF for neutropenia, but only starting from 14 days post-CAR T infusion [58], and the National Comprehensive Cancer Network guidelines recommended against the routine use of G-CSF within 14 days after CAR-T infusion [59]. We support these recommendations and suggest avoiding early use of G-CSF after CAR T.

Thrombopoietin (TPO) agonists are another class of paracrine factors for managing cytopenias. A group reported that TPO administration helped elevate platelet counts and hemoglobin levels while inducing transfusion independence in patients with prolonged grade 3 anemia and grade 4 thrombocytopenia after receiving CAR T [60]. In a single center study, Beyar-Katz et al. reported that all six patients experiencing prolonged severe cytopenia after receiving tisa-cel (*n* = 4) or axi-cel (*n* = 2) who were treated with TPO receptor agonists responded with transfusion independence and resolution of severe neutropenia (ANC > 500/microL) within a median of 22 days [61]. Complete resolution of ANC, platelets, and hemoglobin was observed in 5/6 patients, suggesting that TPO may serve as a useful agent for treating persistent post-infusion cytopenias.

Alternatively, attempts to manage inflammatory toxicities may have a secondary effect of improving cytopenias, such as through blocking IL-1 with agents such as Anakinra (NCT04432506, NCT04150913, NCT04359784). CAR T-cell therapy is known to trigger IL-1 release, which worsens inflammation and activates IL-33 in mast cells [62]. These trials aim to evaluate the dose-dependent impact of IL-1 blockage on treatment related toxicity and prevent or treat inflammatory toxicities such as CRS, ICANS, and HLH. For reasons previously discussed, decreasing inflammatory toxicity may also ameliorate cytopenias. In one single center, retrospective study of patients receiving anakinra for the management of steroid refractory ICANS after treatment with tisa-cel or axi-cel, statistically significant reductions in inflammatory serum cytokine levels with no significant impact on neurotoxicity have been reported [60]. While these results are encouraging, further study is warranted.

Interestingly, GM-CSF seems to be a major driver of CAR T inflammatory toxicity and there are ongoing clinical trials to inhibit its activity in this context [63]. While reducing inflammatory toxicity can be beneficial for HSPC health, deprivation of GM-CSF could be harmful due to lack of myeloid progenitor stimulation. How these two influences balance out and affect cytopenias will be answered in these ongoing trials.

### 6.2. Allogeneic Hematopoietic Cell Transplant following CAR T-Cell Therapy

One strategy to manage post infusion cytopenia is allogeneic hematopoietic cell transplant (allo-HCT) after CAR T-cell therapy. The main reason allo-HCT is generally offered is to increase the durability of remission, not to treat cytopenias. However, an added benefit of the procedure is the treatment of cytopenias.

Clinical outcomes from patients receiving allo-HCT following CAR T therapy are varied. In one systematic review, 17 adult ALL patients receiving an allo-HCT following treatment with a CD19 CAR experienced improvement in event-free survival compared to the patients in CR who did not receive an allo-HCT [64]. On top of this, the transplant related mortality in this group of allo-HCT patients was high at 35% [64]. Another study in pediatric patients receiving a CD28-based, short-lived CAR reported better results [65]. Two out of 21 patients receiving allo-HCT following CAR T relapsed, compared to the six out of seven patients without an allo-HCT [65]. In another pediatric study, 11 out of 40 patients receiving a 41BB-based CAR had an allo-HCT following treatment, and two out of those 11 patients relapsed with CD19+ ALL [66]. Sixteen out of 29 patients in CR who did not receive a transplant relapsed in this study [66,67]. In the ZUMA-3 study, treatment of 10 patients (18%) with an allo-SCT at median time of 98 days following KTE-X19 (brexucabtagene autoleucel) infusion did not significantly increase the median duration of remission [67].

In attempts to further investigate these mixed results and to uncover predictive criteria to identify patients who are more likely to benefit from allo-HCT post CAR T, researchers found that patients who had a shorter median durability of CAR T cells in vivo and without previous transplant history experienced more benefit from an allo-HCT [68]. Overall, more patients receiving shorter-lived CAR T constructs seem to be referred for an allo-HCT, but the benefits from this treatment vary.

Given the superior results in younger patients, age and bone marrow health may play a significant role in their response to transplant post CAR T. In a phase I/II trial, 18/45 patients who were in an MRD-negative CR after receiving CD19 CAR T for B-ALL received an allo-HCT and experienced a higher probability of EFS and OS compared to the other patients with high risk of relapse who did not receive an allo-HCT [32]. After controlling for pre-lymphodepletion LDH concentration and platelet counts, Hay et al. reported that allo-HCT was associated with longer EFS compared to no allo-HCT [32].

As with most retrospective allo-HCT studies, these data are confounded by the fact that the group of patients receiving transplants is, in general, more fit, and healthy. This makes comparison difficult, especially in small studies. Additionally, although the toxicity of allo-HCT is improving, the danger of graft vs. host disease, infections, toxicity from the conditioning regimen, and other drawbacks make this an unattractive approach as a standard option to mitigate cytopenias. As shown in Table 1 and Table 2, incidences of severe or prolonged cytopenias are common across primary studies with different patient populations and CAR T products. 

### 6.3. Autologous Hematopoietic Stem Cell Boost Post CAR T

Another strategy for mitigating post-infusion cytopenias is an autologous hematopoietic stem cell boost (HSCB) following CAR T. This requires prospectively collecting and storing autologous HSCs. In one study, 84% of patients with sustained neutropenia post CAR T (median 43 days) who were treated with an HSCB experienced neutrophil recovery (ANC of >0.5 × 109/L for three consecutive days) or improvement (ANC of >1.5 × 109/L). Importantly, the time to neutrophil recovery post-HSCB was significantly associated with the duration of previous neutropenia and the time between the prior neutropenic episode and HSCB. Moreover, earlier intervention with HSCB was significantly associated with quicker neutrophil recovery [69]. Rejeski et al. reported on the safety and feasibility of HSCB as a salvage therapy for severe hematotoxicity post CAR T-cell therapy and found that HSCB resolved cytopenias in most cases. Out of 13 patients, 11 recovered neutrophils and nine recovered platelets by 30 days post- HSCB infusion [70]. These results emphasize the importance of hematopoietic recovery post CAR T as it relates to the safety and toxicity of the CAR T regimen. Clinical trials are currently investigating ways to integrate the infusion of autologous hematopoietic stem cells (HSCs) into the current CAR T regimen to prevent or treat post-infusion cytopenias.

### 6.4. Alternative Conditioning Regimens

Improved CAR T expansion is associated with the dose intensity of pre-infusion lymphodepleting chemotherapy [71]. It is well established that conditioning chemotherapy improves engraftment of genetically modified cells, presumably by creating space in niches, and providing a more favorable cytokine milieu by reducing competition for lymphopoietic cytokines [72]. Another mechanism which might play a role is the depletion of immunosuppressive cells, such as T regulatory cells and myeloid derived suppressor cells [73].

Cytotoxic chemotherapeutic agents often induce cytopenias, however, prolonged post-infusion cytopenias cannot be explained by lymphodepleting agents alone [74]. This finding suggests that some other factors must contribute to prolonged cytopenias, and perhaps points to the importance of bone marrow microenvironment and immune cell involvement on hematopoietic recovery post CAR T. Strati et al. evaluated the impact of conditioning chemotherapy on lymphocyte kinetics and outcomes in patients with DLBCL and found that the higher change in absolute lymphocyte count (DlIx) from the day of conditioning chemotherapy to the day of axi-cel infusion was significantly correlated with CR [75]. They also found that genetic variations in drug metabolism genes (ABCB1, MISP, and CPVL) were independently associated with DlIx chemotherapy metabolism and CAR T efficacy. This study suggests the importance of underlying genetics on CAR T response and may point towards the benefits of using a more personalized conditioning chemotherapy dosing strategy.

Safe and optimal conditioning chemotherapy dosing is a crucial component of CAR T therapy. Fludarabine is a lymphodepleting agent currently used in all FDA approved CAR T regimens, known for its potency and neurotoxic effects when used at high doses. For CAR T patients, Flu/Cy has been used over other chemotherapies such as low dose Busulfan because it is a successful lymphodepletion agent and, at high doses, is associated with better overall response and CR rates [76]. Busulfan is routinely used as a conditioning agent in pediatric gene therapy because of its reduced toxicity and ability to rid the bone marrow blood cells prior to therapy [77].

Hirayama et al. report that increasing the intensity of Flu/Cy is linked to a more favorable cytokine profile including a higher day 0 monocyte chemoattractant protein-1 and peak interleukin-7 concentrations which are also associated with better PFS [76]. Other groups attempted to reduce the combined Flu/Cy dose and reported favorable results. Kochenderfer et al. treated 22 aggressive B-cell lymphoma patients with a 300–500 mg/m^2^ dose of Cyclophosphomide (Cy) and 125 mg/m^2^ of Flu over three days and noted adequate lymphodepletion with less hematologic and non-hematologic toxicity than higher doses. Overall response rates in this study were 68% [78]. Regardless, these studies highlight the potential for toxicity in the current CAR T regimen and help demonstrate the need for use of better and more targeted chemotherapy agents. Moreover, the finding that pre-infusion cytopenias relate to reduced CAR T efficacy underscores the need to improve conditioning regimens without increasing CAR T toxicity or reducing efficacy.

The combination of Flu/Cy is also important, as the Fred Hutchinson Cancer Research Center used Cy-based lymphodepletion without Flu as well as Flu/Cy in combination and found that the addition of Flu minimized transgene immunogenicity and improved CAR T expansion and persistence in patients receiving CAR T for B-ALL [79]. Hay et al. also found that incorporation of Flu into the lymphodepleting conditioning regimen was associated with better EFS in patients receiving CD19 CAR T for B-ALL [32]. Similar results were observed by Dekker et al. in their retrospective analysis of 26 children and young adult patients receiving tisa-cel for r/r B-ALL, where they found that cumulative fludarabine AUC_T0−∞_ ≥ 14 mg·h/L was correlated with improved LFS [80]. Finally, Gardner et al. reported that addition of Flu was associated with improved CAR T persistence [81]. Combined, these results suggest that optimal Flu dosing may be an important consideration for predicting CAR T durability and efficacy.

Other potential conditioning agents include bendamustine (BEN), an alkylating agent with immunomodulatory properties that is commonly used to treat several hematological malignancies [82]. Specifically, BEN has been shown to suppress myeloid derived suppressor cells, enhance FLT3 expression on dendritic cells, increase production of IL-10 by B cells, inhibit STAT3 activation and suppression of T and B cell proliferation in murine models [82]. In the phase II JULIET trial, investigators were given the choice between using Flu/Cy or BEN as a conditioning agent prior to CART19 infusion. Out of eight patients with NHL and secondary CNS lymphoma treated with BEN as a lymphodepleting agent in the JULIET trial, the ORR was 50% while the CR was 25%. There were no incidences of grade > 2 CRS or ICANS [83]. In a single-center, retrospective analysis of 24 r/r lymphoma patients receiving BEN prior to CAR T in the JULIET trial, the three-month OS was 46%, the CR was 38%, and the three-month PFS was 52%. Incidence of CRS was 29% and ICANS 1% [83]. Across these studies, it appears that the safety of BEN as a conditioning agent is like that of Flu/Cy [83]. In another study, BEN was associated with decreased incidence of CRS, ICANS, and hematotoxicity compared to Flu/Cy in a group of patients treated with CAR T-cell therapy for B-cell lymphoma [84].

Substitutes for Flu/Cy conditioning such as a monoclonal antibody (mAb) to the pan-leukocyte surface protein CD45 have been explored. This mAb can bind to most nucleated hematopoietic cells including myeloblasts and myeloid leukemia cells expressing the hematopoietic-cell-restricted CD45 receptor, but not other non-hematopoietic cell types [85]. Since the anti-CD45 mAb spares other non-hematopoietic cell types from cytotoxicity which are typically killed by non-specific chemotherapy agents, this targeted approach should be significantly less toxic to epithelial cells. Palchaudhuri et al. [86] investigated the use of an internalizing immunotoxin against CD45, called CD45-saporin (SAP), as an alternative conditioning agent for transplant in immunocompetent mice. They found that they could engraft over 90% of donor cells and fully correct sickle-cell anemia in their diseased mouse model. To assess the impact of CD45-SAP on adaptive and innate immunity, they analyzed myeloid recovery in mice receiving CD45-SAP or irradiation by collecting peripheral blood samples at several time points post-conditioning. Circulating myeloid cells reached normal levels by day 12 in the CD45-SAP treated group compared to day 28 in the irradiated group. They also challenged each group of mice with *Candida albicans* two days after conditioning and reported that 100% of the irradiated mice died within 3 days while the CD45-SAP treated mice had an overall survival of 50 days, comparable to untreated control mice. Adaptive immunity was restored, as CD45-SAP treated mice recovered 80% of B cells and 70% of T cells within 18 and 12 days, respectively. Of note, this T cell recovery could have been due to the lack of thymic toxicity with CD45-SAP conditioning compared to irradiation. In all, compared to irradiation with 5Gy TBI, using CD45-SAP prevented neutropenia, anemia, destruction of bone marrow and thymic niches, loss of anti-fungal immunity, and allowed for a quick recovery of T and B lymphocytes.

Together, these results highlight an important feature of targeted therapy: rapid restoration of immunity. Given the relatively long half-life of antibody-based therapeutics, CAR T cells would benefit from modifications conferring resistance to antibody therapy by knocking out, knocking down, or editing CD45. CD45 can fine-tune T cell receptor activity in response to antigen presentation in T cells [87], a function that might not be important in the context of CAR T, making knockout an attractive approach. Indeed, other evidence suggests that the CD45 tyrosine phosphatase present in the immune synapse inhibits CAR T cell activation [88]. Although base editors are also being used to make CD45 invisible to antibodies and CD45-targeted CAR T cells [89].

Attempts at improving the safety and efficacy of the CAR T conditioning regimen are hampered by our superficial understanding of the mechanisms responsible for the benefit of chemotherapy. Replacing chemotherapy will require a more accurate picture of the benefits conferred by fludarabine and cyclophosphamide. Depletion of endogenous T cells may be important, but depletion of normal B cells to improve effector to target ratio may play a role. Depletion of myeloid derived suppressor cells or regulatory T cells could also play a role.

## 7. Alternative Approaches to Optimizing CAR T-Cell Therapy Regimens

### 7.1. Repeat Dosing

Repeat infusion of CAR T-cells (CART2) has been explored to augment therapeutic response. Limited analyses of B-ALL, CLL, and NHL demonstrate modest complete response rates of 19–27% to the second dose [90,91]. Analyses of these cohorts suggest that increased intensity of lymphodepleting chemotherapy correlate with higher CART2 expansion and better response rates [90,91]. While baseline or treatment-induced cytopenias and their effect on outcomes have not been reported in this context, they should be considered in future analyses since patients will be undergoing multiple chemotherapy cycles plus CAR T. Importantly, any potential benefit of repeated dosing relies on recovery of hematopoiesis to clinically meaningful levels. Without that, the risk of a repeat dose of engineered cells with repeated conditioning chemotherapy will not be safe.

Interestingly, in a patient with B-cell lymphoma who relapsed following CD19 CAR T-cell therapy, the CAR T cells seemed to be revived with only a second dose of Flu/Cy. This was included as part of a planned redosing strategy with a second round of CD19 CAR T, but the second dose of cells were never given, only the second round of Flu/Cy [92]. This patient was reported to have severely low circulating CAR T-cells in the peripheral blood five months following CAR T infusion, as confirmed by flow cytometry. Upon treatment with Flu/Cy, pre-existing CAR T-cells revitalized, the patient developed grade 2 CRS, and the relapsed lymphoma partially regressed. These results suggest that some form of maintenance therapy to improve the immunological milieu could be useful for bolstering CAR T persistence and response, assuming the patient has adequate hematopoietic reserve to handle the chemotherapy.

### 7.2. Optimizing the CAR T-Cell Infusion Product Composition

Studies [2,93] have shown that the composition of the CAR T product itself can impact its expansion in vivo and thus its interaction with host immune cells and microenvironment. Single-cell multiplexed cytokine profiling and cellular indexing of transcriptomes and epitopes by sequencing (CITE-seq) analysis of activated CAR T infusion products show that a larger Th2 subset and increased expression of IL-4, -5, and -13 and the upstream regulator of T cell activation genes, GATA-3, were present in products from B-ALL patients who had CR for over 54 months post-infusion compared to those who had CD19+ relapse-free (RF) disease for a median of 9.6 months [94]. In fact, distinct functional cytokine co-expression modules were associated with CR and RF respectively, supporting the notion that there is a more ideal CAR T pre-infusion product that can perhaps be predicted prior to treatment. Additionally, unstimulated products with a greater ratio of immature, stem cell memory T cells (TSCM) and central memory T cells (TCM) to effector memory T cells (TEM) were found in patients with CR compared to those who were RF, indicating that the presence of T-cell differentiation subsets in early memory states is important for CAR T response [94]. Good et al. reported that patients with increased CAR Treg-cell subsets seven days post CAR T infusion experienced accelerated disease progression at six months and reduced neurotoxicity [95]. This is likely due to the immune suppressive properties of Tregs and their lack of cytotoxic potential.

The composition of the CAR T infusion product may affect various CAR T-cell therapy outcomes differently. Haradhvala et al. used single-cell transcriptome sequencing of 105 pre- and post-treatment peripheral blood mononuclear cells samples from patients with large B cell lymphoma treated with CD19 directed CAR T and found that expansion of proliferative CD8+ memory T cells was associated with a more robust response to tisa-cel compared to the heterogeneous T cell populations associated with response to axi-cel [2]. As a result, it may become important to modify T cell subsets in CAR T-cell infusion products in a therapy specific manner to maximize response and expansion in vivo.

### 7.3. Optimizing the CAR Construct

It is important to consider the relationship between CAR structure and CAR T-cell efficacy in vivo. Specifically, the CAR structure plays a critical role in mediating CAR T-cell exhaustion, which results in decreased anti-cancer activity and durability of CAR T cells [96]. T-cell exhaustion is generally defined as chronic stimulation of T cells by antigens [97]. This chronic stimulation can reduce effector T-cell cytokine secretion and increase expression of inhibitory markers. Like T cells, CAR T cells exhibit characteristics of exhaustion, leading to decreased therapeutic efficacy [98].

Optimizing the CAR construct has been a major focus of CAR T-cell research. One potential solution to minimize CAR T-cell exhaustion is through blocking the programmed death-1 (PD-1) cell surface receptor which, upon sustained expression, is associated with decreased T-cell function. PD-1 inhibits TCR expression and downregulates cyclin-dependent kinases which, in turn, slows cell cycling. Cherkassky et al. reported that PD-1 upregulation in tumor microenvironment attenuated T cell function, while cell intrinsic blockade of PD-1 in CAR T-cells restored effector function [99]. Blockade of PD-1 in CAR T-cells may be a useful strategy for improving durability and response to CAR T-cell therapy in patients.

Co-stimulatory domains are critical components of the CAR design, capable of modulating anti-cancer activity and persistence in vivo [96]. For example, Long et al. developed CAR domains targeting disialoganglioside GD2 on sarcoma, containing an scFv derived from the 14g2a antibody, either a CD28 or 4-1BB endodomain, and a CD3-ζ signaling domain. The CAR with a CD28 stimulatory domain, GD2-28z, was more prone to exhaustion than the CAR with a 4-1BB stimulatory domain, GD2-BBz [100]. Upon tonic GD2-28z signaling in the absence of antigen stimulation, GD2 CAR T-cells engaged in scFv-mediated clustering, eventually leading to exhaustion. GD2.BBZ CAR T-cells were similarly overactivated in vivo, but with decreased expression of exhaustion markers, increased cytokine production and better persistence. Overall, the authors found that CD28 domains can alter exhaustion while 4-1BB domains may prevent exhaustion via a specific signaling mechanism. Other researchers have attempted to refine the design of co-stimulatory domains, such as Guedan et al. who developed the ICOS co-stimulator by altering a single amino acid in CD28 [101]. ICOS co-stimulation improved persistence of CAR T-cells, potentially resulting from decreased CAR T-cell differentiation with Th17 cell skewing. Taken together, these studies highlight the important nuances of CAR T-cell design, and how co-stimulatory domains play a key role in CAR T-cell exhaustion. Optimizing the CAR construct is an important step in improving response to CAR T.

## 8. CAR T and Immune Cell Interaction

Preclinical data suggest that CAR T cells interact with macrophages and other immune cells, and this interaction may reduce CAR T toxicity while improving efficacy [37]. In one study [37], immune-deficient transgenic mice (NSG-S) injected with human CD34+ cells experienced greater reduction in tumor burden compared to their immune-incompetent counterparts, suggesting that normal hematopoiesis may provide some clinical benefit with CAR T response in vivo. In another experiment from that same study, NSG-S mice that retained myeloid cells after receiving CD33 KO CD34+ cells and a CD33 directed CAR, experienced greater reduction in tumor burden and less toxicity compared to mice that loss myeloid cells after receiving control CD34+ cells [37]. Together, these results suggest that myeloid cells may support CAR T cells in vivo. With that, there are some confounding variables in this study; namely, the use of CD33 KO CD34+ with a CD33 directed CAR for the experimental group. This design allowed the CD33 CAR to target only CD33+ tumor cells without targeting the CD33 marker on healthy myeloid cells. By contrast, the CD33 directed CAR was able to target CD33+ tumor cells and normal myeloid cells. This competition may have increased CAR T related toxicity and reduced therapeutic efficacy because the CAR was targeting both healthy and diseased CD33+ cells.

## 9. CAR T and Clonal Hematopoiesis

Clonal hematopoiesis (CH) commonly occurs in patients with hematologic malignancies and may affect CAR T induced inflammatory toxicities such as CRS and ICANS [102]. Clonal hematopoiesis is defined as an accumulation of a single mutated hematopoietic stem cell. Certain clones have the potential to become malignant, and several hematologic malignancies have been associated with somatic mutations of CH-associated genes such as DNMT3A, TET2, TP53, and ASXL1. Due to the nature of CAR T being a later-line treatment for hematologic malignancies, the incidence of CH-associated genes is higher in patients receiving CAR T than the general population [103,104].

In a study of 114 patients (median age 63 years) with B-cell lymphoma who received CD19 CAR T, somatic mutations were found in the peripheral blood samples of 36.8% of patients [105]. Of note, increased severity of CRS and ICANS was seen in patients with the DNMT3A and TET2 somatic mutations. CH did not seem to impact CAR T efficacy or survival rates in patients with CH compared to patients without [106]. However, CH with somatic mutations in specific genes have been linked to increased IL-1 and IL-6 levels which are key drivers of CRS and ICANS [107]. There could be mutations which lead to increased CAR T-associated toxicity [102].

CH may play an impactful role on the development of CAR T-cells considering they are derived from the patients’ own cells which have been exposed to many cycles of cytotoxic therapy and years of age-related inflammation [102]. For example, T-cells derived from a patient that harbors CH-associated genes can expand ex-vivo and potentially outcompete other T-cells in culture, leading to a change in the CAR T product and potentially its efficacy in vivo [102]. Interestingly, Fraietta et al. reported that disruption of CH-associated genes TET2 and CBL during CAR T manufacturing led to cell-intrinsic enhancement of CAR T-cell expansion in culture and achievement of CR after a history of non-responsiveness in a patient with chronic lymphocytic leukemia [107]. Specifically, the TET2 deficient clones exhibited central memory T-cell properties which led to greater T-cell expansion in vivo, persistence, and anti-cancer activity [107]. In a similar vein, a preclinical study reported that DNMT3A deletion in CAR T cells was protective against CAR T-cell exhaustion and improved anti-canter activity in animal models [108]. It is possible that disruption of certain CH-associated genes in the CAR T product may improve durability and response in vivo, though the development of a CAR T-associated malignant clone is a risk.

The significant heterogeneity in the patient population limits these studies. This heterogeneity is partly due to the various genes that can cause CH, but also the extent to which the T cells themselves harbor the CH mutation. Mutated HSC clones tend to have some myeloid skewing and T cells can be long-lived and formed from non-CH HSCs [103]. Future studies can clarify how these factors impact the CAR T cells in a cell-autonomous and non-autonomous fashion.

## 10. Discussion

CAR T-cell therapy is an exciting and evolving approach to treating hematologic malignancies. However, the benefit to risk ratio can limit its use. Potential approaches to improving the CAR T regimen are being investigated such as modifying the CAR T-infusion product itself, adjusting the conditioning regimen, and mitigating severe side effects. For example, the use of paracrine factors—such as G-CSF, TPO, or cytokine inhibitors—have been used to manage post-infusion cytopenias. Other approaches to address treatment related cytopenias include HSC boost after CAR T infusion. Different groups are evaluating less toxic conditioning agents such as the anti-CD45 MAb. Alternatively, the CAR T product itself can be modified through the manipulation of T cell subsets.

Overall, there are many avenues for improving the current CAR T-cell therapy regimen and several potential applications for CAR T as a platform for individualized medicine. Combinations of the approaches may lead to more effective and tolerable therapy for patients in need.

## Figures and Tables

**Figure 1 cells-12-00531-f001:**
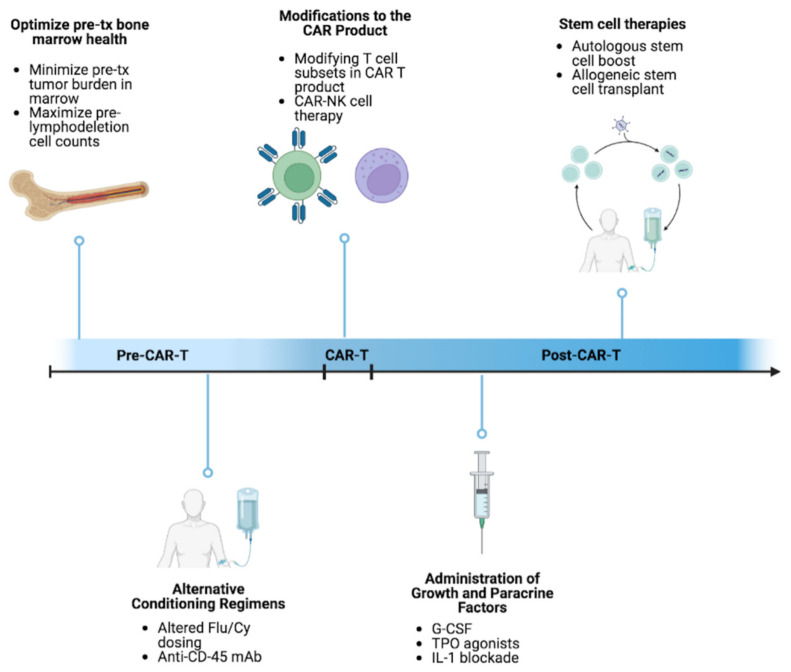
Strategies for improving hematopoiesis in the CAR T-cell setting to augment therapeutic response.

**Table 1 cells-12-00531-t001:** Incidence of Prolonged Cytopenia in a Sample of CAR T Primary Studies.

Reference	Sample Size	Disease	Study	CAR Construct	Incidence of Prolonged Persistent Cytopenia
Locke et al. [4]	119	DLBCL, PMBCL, t FL	ZUMA-1 (phase I/II)	Anti-CD19, CD28 co-stimulatory domain (retroviral)	7%, 11%, (>3 mo)
Schuster SJ et al. [8]	167	DLBCL	JULIET (phase II)	Anti-CD19, 4-1BB co-stimulatory domain (lentiviral)	32%, (>1 mo)
Wang et al. [40]	74	MCL	KTE-X19 (phase II)	Anti-CD19, CD28 co-stimulatory domain (retroviral)	16%, 16%, (>3 mo)
Maude et al. [48]	75	B-ALL (children and young adults)	ELIANA (phase I/II)	Anti-CD19, 4-1BB co-stimulatory domain (lentiviral)	12%, 53%, (>1 mo)
Lee et al. [65]	53	B-ALL (children)	NCT01044069 (phase I)	Anti-CD19, 4-1BB co-stimulatory domain (lentiviral)	33%, (>14 days)

DLBCL—Diffuse Large B-cell Lymphoma; t FL—Transformed Follicular Lymphoma; PMBCL—Primary Mediastinal B-cell Lymphoma; B-ALL—Acute Lymphoblastic Lymphoma.

**Table 2 cells-12-00531-t002:** Clinical Studies and >Grade 2 Toxicities.

Reference	Study	CAR Construct	Anemia	Thrombocytopenia	Neutropenia	CRS	ICANS	Infections
Locke et al. [4]	ZUMA-1 (phase I/II)	Anti-CD19, CD28 co-stimulatory domain (retroviral)	43%	38%	78%	13%	28%	8%
Schuster SJ et al. [8]	JULIET (phase II)	Anti-CD19, 4-1BB co-stimulatory domain (lentiviral)	39%	28%	33%	22%	12%	20%
Wang et al. [40]	KTE-X19 (phase II)	Anti-CD19, CD28 co-stimulatory domain (retroviral)	50%	51%	85%	15%	31%	32%
Maude et al. [48]	ELIANA (phase I/II)	Anti-CD19, 4-1BB co-stimulatory domain (lentiviral)	Not reported	41%	35%	46%	13%	24%
Lee et al. [65]	NCT01044069 (phase I)	Anti-CD19, 4-1BB co-stimulatory domain (lentiviral)	68%	53%	>50%	40%	0%	Not reported
Shah B.D. et al. [67]	ZUMA-3 (phase II)	Anti-CD19, CD28 co-stimulatory domain (retroviral)	49%	30%	27%	24%	25%	25%

DLBCL—Diffuse Large B-cell Lymphoma; t FL—Transformed Follicular Lymphoma; PMBCL—Primary Mediastinal B-cell Lymphoma; B-ALL—Acute Lymphoblastic Lymphoma.

## Data Availability

All data presented herein is available to the public via the sources cited.

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
