# Peer review of "Chimeric Antigen Receptor T-Cell Therapy and Hematopoiesis"

_cells, 2023, doi:10.3390/cells12040531_

Round 1
Reviewer 1 Report
The review manuscript aimed at giving an overview of the influence on hematopoiesis by Chimeric Antigen Receptor (CAR) T-cell therapy. The authors mainly summarized the incidence and characteristics of CAR T-associated cytopenias, factors associated with post-CAR T cytopenias, pre-treatment cytopenias and reduced CAR T efficacy, implications of age-associated inflammation on bone marrow and CAR T outcomes, approaches to mitigating and managing cytopenias, potential alternative approaches to optimize CAR T-cell therapies, and mechanism of interaction among CAR T, immune cells, and hematopoiesis system. The manuscript concluded current strategies to treat or avoid CAR T -cell related cytopenias. This information would be very useful for developing further CAR T-cell therapies and therapies to cytopenia.
The manuscript is clear, and relevant for the field, but the structure can be improved. The authors mentioned the potential mechanism of cytopenia at the end of the manuscript. It would be easier for the reader to understand if this part can be arranged to the beginning with supplementing some background information of cytopenia.
The manuscript is very informative and the language is clear. However, it would be better if the authors can make more comments in each section. In addition, there are some small formatting mistakes in the manuscript.
In line 140, there are two “x 103”, the author might mean “x 10^3”.
In line 155, it would be easier for the readers to understand if using phrase “following CAR T therapy” than “following CAR T”.
In line 165, the abbreviation “B-NHL” was used without previous full name mentioned.
In line 213, the abbreviation “MRD” was without previous full name mentioned.
In line 645, citation 101 was mentioned. However, there is no 101 in the reference list. The references ranged from 0 to 100, but reference 0 was not mentioned in the article. It seems there is a numbering mistake.
It would be better if the author can synchronize the format of the indent. There are spaces at the beginning of most paragraphs. However, the spaces are missing in line 244, 304, 432, 543, 566, 599, 617, 657, and 666.
Author Response
We thank Reviewer 1 for his or her helpful comments. We have now moved a section with an overview of the mechanism of cytopenias into the third paragraph of the introduction. We have also made more comments throughout to give more color and explanation to the data. We did our best to address the formatting mistakes, typos, abbreviations, and citations in the previous version of the manuscript.
Reviewer 2 Report
This is a very well written review that highlights all the implications of the tumor microenvironment (i.e. the bone marrow niche) and the pre- and post- CAR T treatments in the overall efficacy and safety of CAR T cell therapy.
The authors have raised important questions that surely help in guiding for better design of the clinical studies with CAR T cells, suggesting some solutions that are context-related and leaving some interesting open questions that still need to be solved in the next future.
Minor comments:
- Abstract: line 13, I would change "epithelial cancers" with the more general "solid cancers".
- Since in the paragraph 2 the authors suggest the CAR product to play a role in the risk of cytopenia, I would add a column in the tables I and II describing the CAR construct (generation/costimuli, lenti/retro/non-viral, etc)
- line 81-82 wrong editing "duation" and "durtion" instead of duration
- line 233 wrong reference. 35 instead of 36.
Author Response
We thank Reviewer 2 for his or her helpful comments. We changed "epithelial cancers" to the more general "solid cancers". We also added a column to tables I and II describing the CAR construct. We fixed the typo in line 81-82 and the reference in line 233.
Reviewer 3 Report
Bryanna Reinhardt and colleagues present a quality and well-written review manuscript focused on chimeric antigen receptor T-cell therapy and hematopoiesis.
Authors summarized the characteristics and risk factors associated with post CAR T-cell cytopenias, as well as the current evidence linking cytopenias and therapeutic response. They then delve into ongoing avenues of investigation to mitigate these cytopenias, including both modifications to the CAR T-cell regimen and adjunctive therapies.
Authors discussed a variety of potential solutions being investigated to address the limitations of CAR T. First, they discuss the incidence and characteristics of CAR T related cytopenias and their association with reduced CAR T-cell efficacy. Authors reviewed approaches to managing or mitigating cytopenias during the CAR T regimen including the use of growth factors, allogeneic rescue, autologous hematopoietic stem cell infusion, and alternative conditioning regimens. Finally, they introduce novel methods to improve CAR T-cell infusion products and the implications of CAR T and clonal hematopoiesis.
Finally, authors conclude that there are many avenues for improving the current CAR T-cell therapy regimen and several potential applications for CAR T as a platform for individualized medicine. Combinations of the approaches may lead to more effective and tolerable therapy for patients in need.
Overall, the manuscript is highly valuable for the scientific community and should be accepted for publication after minor edits are made.
=======================================
Other comments:
1) Please check for typos and punctuation throughout the manuscript.
2) Authors are kindly encouraged to cite the following article that overviews various aspects of CAR-T cell immunotherapy.
DOI: 10.3390/cancers14041078
Author Response
We thank Reviewer 3 for the helpful comments and recommendation. We have checked for typos and punctuation throughout the manuscript and did our best to correct them. We also added the excellent citation.